# Exploring multisite heterogeneity of human basal cell carcinoma proteome and transcriptome

Ariel Berl[1]☯*, Ofir Shir-az[1]☯, Ilai Genish[2]☯, Hadas Biran[3], Din Mann[1], Amrita Singh[4], Julia Wise[4], Vladimir Kravtsov[5], Debora Kidron[5], Alexander Golberg[4]*, Edward Vitkin[2], Zohar Yakhini[2,3]*, Avshalom Shalom[1]

1 Department of Plastic Surgery, Meir Medical Center, Kfar Sava, Israel, Affiliated with the Faculty of Medicine, Tel Aviv University, Tel Aviv, Israel, 2 Efi Arazi School of Computer Science, Reichman University, Herzliya, Israel, 3 Department of Computer Science, Technion - Israel Institute of Technology, Haifa, Israel, 4 Department of Environmental Studies, Porter School of Environment and Earth Sciences, Tel Aviv University, Tel Aviv, Israel, 5 Department of Pathology, Meir Medical Center, Kfar Sava, Israel, Affiliated with the Faculty of Medicine, Tel Aviv University, Tel Aviv, Israel

☯ These authors contributed equally to this work.
* arielberl23@gmail.com (AB); agolberg@tauex.tau.ac.il (AG); zohar.yakhini@idc.ac.il (ZY)

**Data Availability Statement:** All relevant data are within the paper and its Supporting information files. Datasets related to this article can be found at

## Abstract

Basal cell carcinoma (BCC) is the most common type of skin cancer. Due to multiple, potential underlying molecular tumor aberrations, clinical treatment protocols are not well-defined. This study presents multisite molecular heterogeneity profiles of human BCC based on RNA and proteome profiling. Three areas from lesions excised from 9 patients were analyzed. The focus was gene expression profiles based on proteome and RNA measurements of intra-tumor heterogeneity from the same patient and inter-tumor heterogeneity in nodular, infiltrative, and superficial BCC tumor subtypes from different patients. We observed significant overlap in intra- and inter-tumor variability of proteome and RNA expression profiles, showing significant multisite heterogeneity of protein expression in the BCC tumors. Inter-subtype analysis has also identified unique proteins for each BCC subtype. This profiling leads to a deeper understanding of BCC molecular heterogeneity and potentially contributes to developing new sampling tools for personalized diagnostics therapeutic approaches to BCC.

## Introduction

Skin cancers are the most prevalent type of malignancy worldwide, with a substantial impact and burden on public health [1–7]. Basal cell carcinoma (BCC) is the most common type of skin cancer [1–3]. In sporadic cases of skin cancer, sun exposure is the major risk-factor for induction of tumorigenesis. Originating from cells in the basal cell layer of the epidermis and around hair follicles, the main histological subtypes of BCC are nodular, micronodular, superficial, morpheaform, infiltrative and fibroepithelial [8]. BCC rarely metastasizes or causes death; yet, it leads to extensive morbidity due to local tissue invasion and destruction [9, 10]. Surgical excision is the preferred method for treating BCCs and when not feasible, other

https://github.com/GolbergLab/BCC_
Heterogeneity,hostedatGithub.

**Funding:** Israel Research Authority, Kamin
Program Golberg, Shalom, Yakhini Israel Ministry
of Science and Technology Golberg, Yakhini
EuroNanoMed3 MATISSE Project Golberg, Shalom
SPARK-TAU Golberg, Shalom, Yakhini TAU Zimin
Institute for Engineering Solutions Advancing
Better Lives Golberg, Shalom The funders had no
role in study design, data collection and analysis,
decision to publish, or preparation of the
manuscript.

**Competing interests:** The authors have declared
that no competing interests exist.

modalities such as cryotherapy, radiation and targeted molecular therapy can be implemented [11].

Molecular heterogeneity of tumors and of BCC specifically, is a well-known phenomenon and presents a challenge when considering diagnosis and treatment options [12, 13]. Analysis of genetic material, transcribed genes, metabolites and proteins can add important information in the quest to understand the molecular malignant pathogenesis and to develop future therapeutic targets [14–16]. Tumor specific genetic alterations are the major focus of the development of novel therapies targeting specific pathways in BCC [12]. Personalized medicine, where the proposed therapy is tailored according to patient characteristics, genetic features and tumor specific properties, aims to offer patients more effective treatment with higher safety profiles. This approach holds promise for improving treatment protocols and results, while reducing costs [17].

The most frequent aberrancy encountered in BCC molecular genetics is a mutation in the *PTCH* gene in the Hedgehog (HH) pathway [18–20]. Therefore, the HH pathway, which is upregulated in BCC, is the current target of several approved molecular inhibition therapies [18–21]. However, previous studies have described other signaling pathways and the heterogeneous nature of BCC [12, 22–28]. A large gap remains between understanding the relevance of these pathways to the pathogenesis of BCC and their therapeutic potential. The molecular diagnosis of BCC has been researched and various markers have been proposed for diagnosing primary BCC or in rare cases metastatic BCC [29]. However, the molecular heterogeneity of these tumors limits the utility of these markers [12]. While current therapies for BCC focus on the HH pathway, additional possibilities and other targets for therapy and diagnosis are yet to be elucidated.

This study investigated the multisite molecular heterogeneity profile of human BCC based on RNA and proteome profiling, using 3 locations in lesions excised from 9 patients (one lesion per patient). We focused on intra-patient (within a tumor excised from a single patient) and inter-patient (between the tumors excised from different patients) heterogeneity of the nodular, infiltrative, and superficial BCC subtypes.

This study adds to the current literature in the quest to elucidate BCC heterogeneity, and compares multisite derived samples both within tumors and between patients.

By using a novel approach to evaluate heterogeneity, our study of genetic and molecular BCC heterogeneity identified differences in the RNA and proteome components of each unique BCC tumor and between the different parts of each lesion. These findings will influence future research for molecular sampling tools and will advance the development of personalized therapy.

## Methods

From March 2020 to March 2021, transcriptomic and proteomic samples were collected from BCC lesions from 9 patients who underwent surgical excision of a skin lesion suspected as BCC, at Meir Medical Center, Israel (Table 1). RNA was extracted from 3 patients (9 samples, 3 per tumor, one tumor per patient) and proteomics were analyzed for 9 patients (27 samples, 3 per tumor, one tumor per patient), as described below. All lesions excised were at least 1 cm in diameter.

### Sample collection

Following excision of the lesion, a strip along the full length of the lesion was excised preserving the deep and lateral margins to prevent obscuring histological analysis. Each sample included both tumor tissue and healthy margins. The strip was then split longitudinally into 2

**Table 1. Patient demographics and lesion characteristics.**

| Patient | Sex | Age | BCC subtype | Lesion location | Proteomics/RNA seq |
|---|---|---|---|---|---|
| 1 | Female | 66 | Infiltrative | Lower eyelid | Proteomics + RNA seq |
| 2 | Female | 90 | Infiltrative | Right chin | Proteomics |
| 3 | Male | 88 | Infiltrative | Left temple | Proteomics |
| 4 | Male | 84 | Superficial | Right cheek | Proteomics + RNA seq |
| 5 | Male | 68 | Superficial | Right shin | Proteomics |
| 6 | Male | 70 | Superficial | Right shoulder | Proteomics |
| 7 | Male | 85 | Nodular | Nose | Proteomics + RNA seq |
| 8 | Female | 76 | Nodular | Upper lip | Proteomics |
| 9 | Male | 80 | Nodular | Right chin | Proteomics |

parts. The orientation of the two pieces was marked and one was sent for molecular profiling as a fresh sample on 0.9% saline-soaked gauze for tissue lysis within 30 minutes of excision. The second sample was sent in formaldehyde for histopathological processing. The first tissue strip was cut into 3 pieces representing 3 locations: 1 from each end and 1 in the center, yielding a total of 27 samples for proteomics and 9 samples for RNA analysis. Patient data and demographics were collected from electronic medical records. Histological characteristics of each lesion were reviewed independently by two independent certified dermatopathologists.

## Isolating total RNA from tissue using lysis buffer

RNA was extracted from cut tumor tissue, similar in orientation to the tumor. The EZ-RNA II kit (Biological Industries, Beit Haemek Ltd., Beit Haemek, Israel) was used for the RNA isolations. (The protocol is further described in Supplement S1 in S1 File. Isolating total RNA from tissue using lysis buffer.)

## RNA sequencing

The concentration and integrity of the RNA were determined and a cDNA was generated (RNA sequencing protocol is further described in Supplement S1. RNA sequencing in S1 File).

## Protein extraction

The samples were taken from a slice of tumor tissue, preserving the orientation of the samples to the original tumor. The EZ-RNA II kit was used for protein isolation. The protein pellets were air dried and taken for proteomic analysis. (Protein sample preparation protocol is further described in Supplement S1. Protein extraction in S1 File).

## Mass spectrometry analysis

Mass spectrometry was performed in a positive mode, using a repetitive full MS scan followed by collision-induced dissociation of the 10 most dominant ions selected from the first MS scan. In addition, one sample, not related to the dataset, was sent through the liquid chromatography-mass spectrometer (LCMS) to measure the LCMS machine "noise". (The mass spectrometry sample preparation and analysis protocol are further described in Supplement S1. Mass spectrometry analysis in S1 File).

## Statistics and reproducibility

A total of 9 RNA and 27 proteomic samples were used in this study. Data analysis was conducted with *python* (ver. 3.7.4) using *pandas* and *scipy* packages. Statistical analysis included the use of a heatmap approach based on Spearman and Pearson correlations. Histograms were created for both the Pearson and Spearman correlations and a one-sided Wilcoxon Rank-Sum test examined the differences between the distributions of correlations within different groups. Inter-sample intersections of the observed tumor transcriptome and proteome were analyzed and are presented using Venn diagrams.

Differential analysis was used to produce a list of genes whose expression was variable intra-patiently. The list of genes was analyzed with the GOrilla web tool to identify overabundant cellular processes.

Possible single nucleotide variations (SNVs) were compared in pairs of intra-patient samples and identified genome positions in which these samples were significantly different. The significance of the identified positions was assessed with a Jensen-Shannon distance score.

## Proteomics and transcriptomic data analysis

Data were analyzed based on *python* (ver. 3.7.4) using *pandas* and *scipy* packages. The visualizations were generated using the *matplotlib* package.

**RNA-seq alignment.**   We used STAR (version 2.7.3a) [30] to process FASTQ files and produce BAM files (for mutation calling) and gene counts (for RNA-seq differential expression analysis). Before the analysis gene expression data were normalized to counts per million (CPM) and genes with less than 0.5 CPM in all samples were filtered out [31].

**Heatmap of the heterogeneity.**   A heatmap approach was used to depict the heterogeneity of the protein samples. In both axes of the heatmap, the sample names are ordered by patients and their BCC subtypes for ease of visualization (same order in both axes). Spearman and Pearson correlation coefficients were calculated for the proteomics profiles based on label-free quantitation protein intensities of each pair of samples. Colors of the heatmap were adjusted to maximize the visual differences in correlation values between the adjacent cells. Spearman and Pearson correlations were used to measure monotonic and linear relationships, respectively.

## Correlations, histograms and distributions

Density histograms were created to summarize the values represented in the heatmaps. This analysis was conducted separately for the Spearman and Pearson values for the protein findings.

The correlations between samples were divided into two groups: the correlations between different patients (i.e., inter-patient and inter-tumor) and the correlations between the samples of the same patient (i.e., intra-patient and intra-tumor), further referred to as Group 1 and Group 2, respectively. Low correlation values within Group 2, comparable to those of Group 1, suggest high tumor heterogeneity. The Bayesian probability of misclassifying the correlation value from Group 2 as sampled from Group 1 was estimated per each value bin separately and then totaled, as described in **Eq. 1** in **Supplement S1 of** S1 File.

This process was repeated both for the Pearson and Spearman correlations. We also used a one-sided Wilcoxon Rank-Sum test examine the differences between the distributions of correlations within different groups.

In addition, we show the distribution of correlations within the noise control group (Group 3), dedicated to evaluating the LC/MS/MS measurement noise and designed as 5 subsequent LCMS measurements of the same sample not related to the dataset.

### Group-intersection heterogeneity analysis

We analyzed the inter-sample intersections of the observed tumor transcriptome and proteome to alternatively quantify the heterogeneity of the gene expression of BCC tumors. This analysis is presented using Venn diagrams and each figure is accompanied by a table showing the sizes of each diagram slice on the patient level (row per each studied patient).

### RNA-seq differential expression analysis

Gene expression data were normalized to counts per million. To retrieve a list of heterogeneous genes among the different locations of the tumor for each patient, we computed the maximum absolute fold change (FC) for all pairwise comparisons of the patient samples for each gene. Only genes that appeared in the lists of at least two patients were considered. The final list consisted of 156 genes (p = 2.10e-152, see **Supplementary material** SO in S1 File [30, 32–35] for computational details and SO for the list of genes). The list was analyzed with the Gorilla web tool [31, 36] to identify overabundant cellular processes.

### Prediction of RNA mutations

To detect possible somatic single nucleotide variations (SNVs) we compared pairs of intrapatient samples and identified genome positions in which these samples were significantly different. The base assumption was that the distribution of expressed alleles should be similar across skin cells of the same individual [31]. The significance of the identified positions was assessed with a Jensen-Shannon (JS) distance score. (Further described in **Supplementary material** SO. RNA analysis in S1 File).

## Results

A total of 9 patients with excised, suspected BCC skin lesions were included in this study. The average patient was 78.5 years old (range 66–90 years) and 6 were male. Histopathologic analysis of the lesions was consistent with the diagnosis of BCC and determined to be 3 cases of nodular, 3 infiltrative and 3 superficial (Table 1).

### Proteomics analysis

In total, 27 samples were collected from 9 patients (3 samples per patient); 3 patients per each histologically-confirmed BCC subtype (infiltrative, superficial, nodular). A total of 5,782 proteins were identified in at least one sample after lysis and proteomic analysis.

Heatmap-based analysis for heterogeneity shows Spearman (upper-right triangle) and Pearson (lower-left triangle) correlations. The 9 x 9 squares with yellow boundaries (excluding 3 x 3 blue-boundary diagonal) represent the correlation values between different patients with similar BCC subtype (Group 1, intra-subtype). The 9 x 9 squares with red boundaries represent the correlations between samples from the different BCC subtypes (Group 1, inter-subtype). The 3 x 3 squares with blue boundaries along the diagonal represent the correlations between 2 samples from the same patient (intra-patient, Group 2) (Fig 1).

Histograms of the observed densities were constructed for the distribution of correlations in Groups 1 and 2. The histograms are displayed in Fig 2 in yellow-red and blue, respectively. Although the correlations in Group 2 are significantly higher than those in Group 1 (Wilcoxon one-sided test p = 3.256e-15), the meaningful intersection between the histograms implies significant heterogeneity of specific tumors, clearly rebutting the hypothesis of lesion proteomic homogeneity. In other words, in notably many cases, the difference between two proteomic samples extracted from the same tumor of a single patient was as significant as the difference

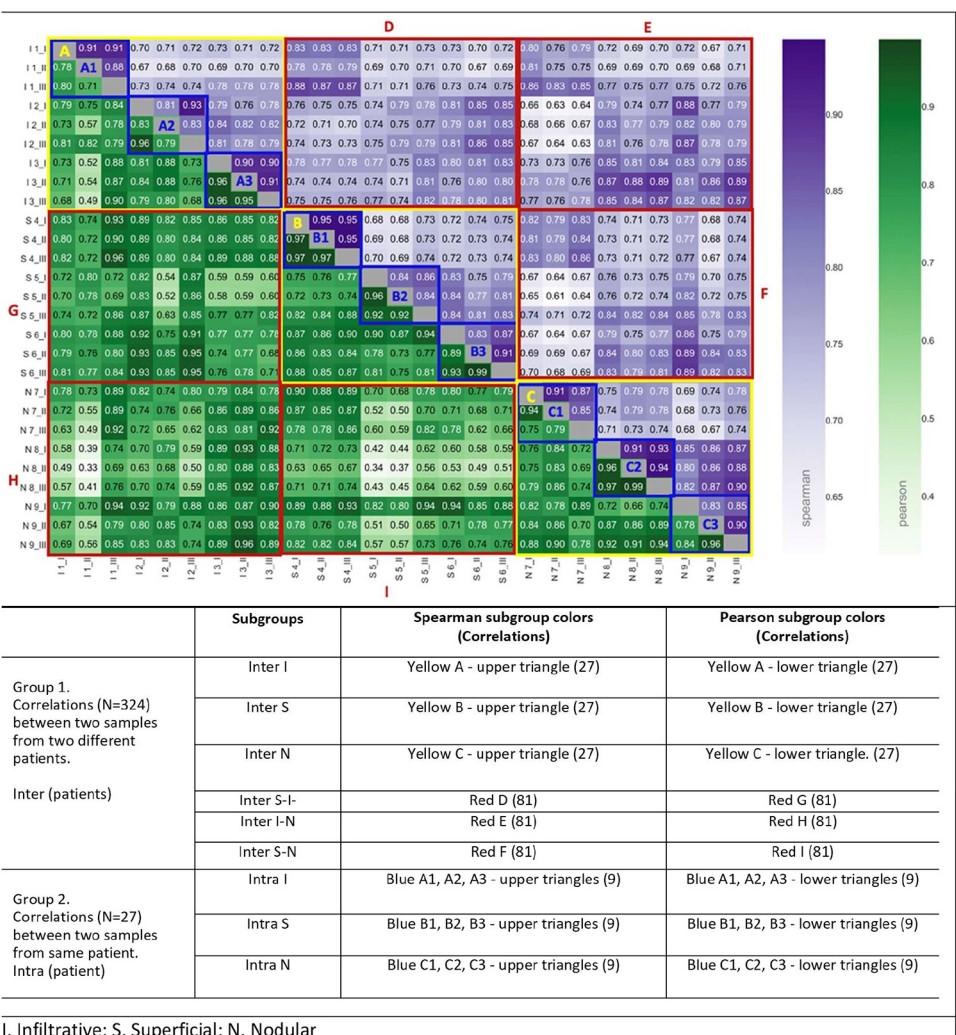

**Fig 1. Proteomic heatmap-based heterogeneity analysis.** The heatmap shows Spearman and Pearson correlations of proteomic profiles based on LFQ-intensity measurements between each two *different* samples from our dataset (overall, 5782 proteins were identified in at least 1 of all 27 samples). The table below the figure describes the groups of empirical correlations and the color coding of blocks in the heatmap (also see text). All correlations were calculated on the LFQ-intensity of values. Interpretation example: The Pearson correlation between the proteomic profile measured in sample from Location I from Patient 1 (Infiltrative subtype) to proteomic profile measured in sample from Location II from Patient 4 (Superficial subtype) is 0.80, while the Spearman correlation for these samples is 0.83.

between two proteomic samples extracted from two different patients, sometimes even from two different BCC subtypes. Quantitively, the Bayesian probability of misclassifying the correlation (Group 1 vs. Group 2) was 40.7% and 48.1% for Spearman and Pearson tests, respectively. Moreover, the observed correlations within Group 3 (grey histogram in Fig 2) representing random measurement noise are significantly higher (Group 3>Group 2 and Group 3>Group 1). The Wilcoxon one-sided test p-value based on Pearson correlations were 1.948e-06 and 3.592e-08, respectively, implying that all observed correlations differ more significantly from each other than expected from the noise of the measurement process.

Group-intersection heterogeneity analysis of the proteomic data (Fig 3) provides additional information on intra-patient heterogeneity. The diagram shows 1760 proteins ("all samples"

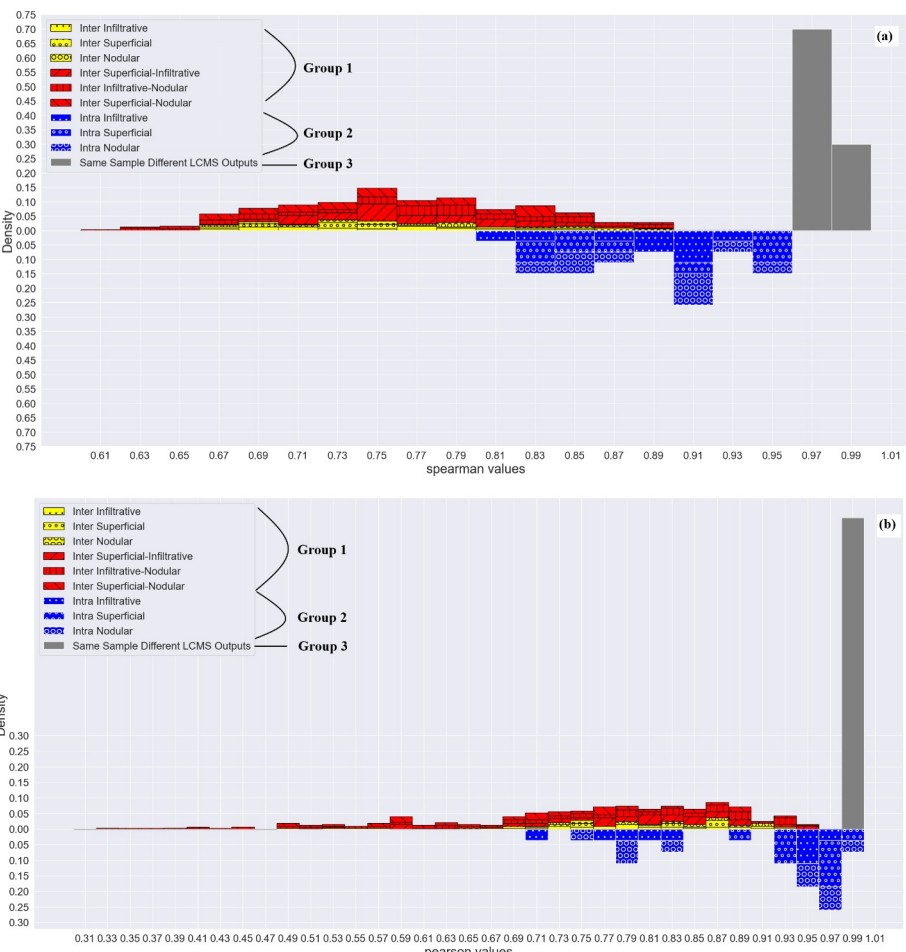

**Fig 2. Group correlations and distribution histograms.** (**a**) Spearman and (**b**) Pearson histograms summarizing the values in the heatmap presented in Fig 4. The values are summarized as normalized, observed densities separately for the inter-patient (Group 1) and intra-patient (Group 2) correlations, displayed respectively as red (upper) and blue (lower) bars. The correlations measured within the noise control group (Group 3) are presented as grey bars.

slice) measured with a positive label-free quantitation intensity in all BCC samples, which is only about 30% of all 5782 observed genes. Per BCC subtype ("all sub-type samples" slice), we observed 2509 proteins shared among infiltrative BCC samples, 2099 among superficial and 2504 among nodular subtypes. The number of uniquely identified proteins per patient sample ranged from 44 to 413 (1–12% of the proteins identified overall in the dataset) per sample (random expectation is 10.6 unique proteins per location), implying high potential heterogeneity in BCC proteomics.

The hypothesis of high heterogeneity in BCC proteomics is also supported by the number of proteins always observed in each BCC subtype with partial multisite profile. We identified 130, 104 and 107 proteins (for Infiltrative, Superficial and Nodular sub-types respectively) that appear in each patient but not in all patient's locations (Table 2).

Analyzing inter-subtype heterogeneity, we identified 11 proteins observed in each Infiltrative patient, but not in Superficial patients; 6 proteins observed in each Infiltrative patient, but not in Nodular patients; 3 proteins observed in each Superficial patient, but not in Nodular patients; 4 proteins observed in each Nodular patient, but not in Infiltrative patients; and 14

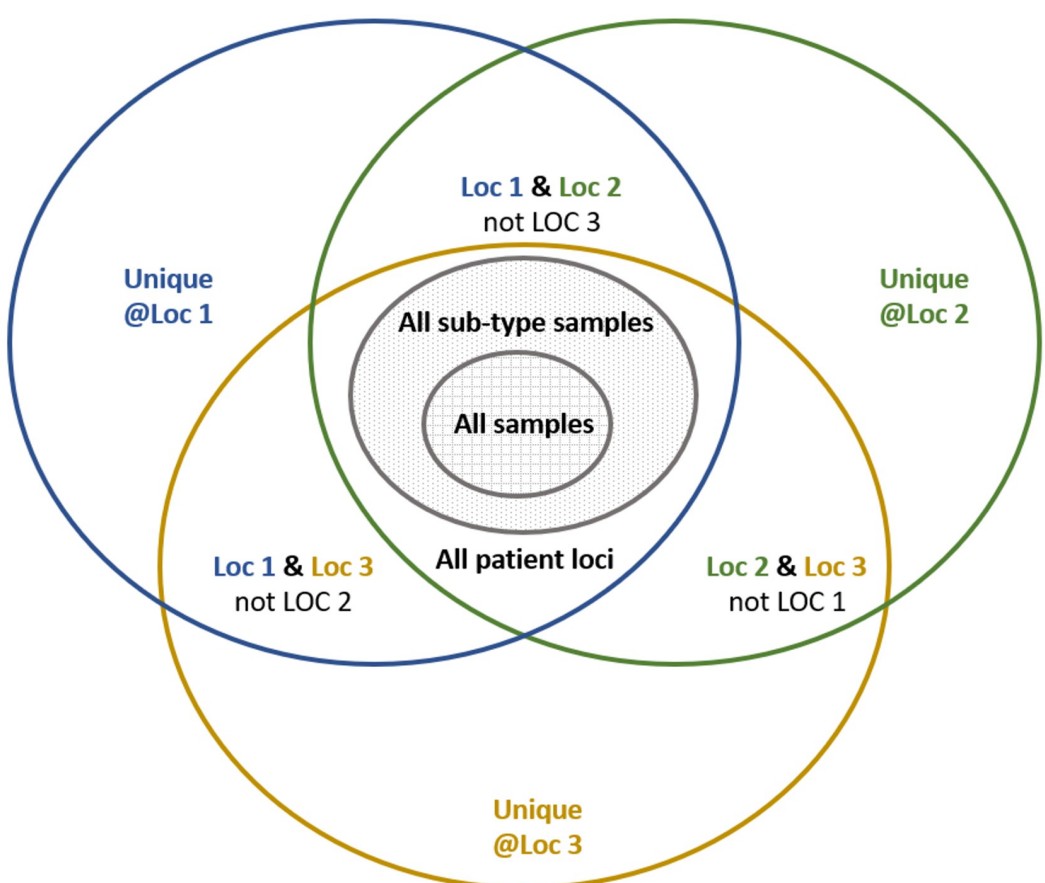

| | SubType | All samples | All sub-type samples | All patient loci | Loc 1 & Loc 2 not Loc 3 | Loc 1 & Loc 3 not Loc 2 | Loc 2 & Loc 3 not Loc 1 | Unique @Loc 1 | Unique @Loc 2 | Unique @Loc 3 |
|---|---|---|---|---|---|---|---|---|---|---|
| | | | | Genes shared among | | | | Genes unique to | | |
| Patient 1 | Infiltrative | 1,760 | 2,509 (+749) | 3,200 (+691) | 3,318 (+118) | 3,453 (+253) | 3,387 (+187) | 85 (2% of 3,656) | 95 (3% of 3,600) | 244 (6% of 3,884) |
| Patient 2 | Infiltrative | | | 3,439 (+930) | 3,592 (+153) | 3,832 (+393) | 3,585 (+146) | 150 (4% of 4,135) | 252 (6% of 3,990) | 102 (3% of 4,080) |
| Patient 3 | | | | 3,468 (+959) | 3,700 (+232) | 3,560 (+92) | 3,743 (+275) | 89 (2% of 3,881) | 265 (6% of 4,240) | 122 (3% of 3,957) |
| Patient 4 | Superficial | | 2,099 (+339) | 3,709 (+1,610) | 3,884 (+175) | 3,850 (+141) | 3,789 (+80) | 104 (3% of 4,129) | 62 (2% of 4,026) | 44 (1% of 3,974) |
| Patient 5 | Superficial | | | 2,450 (+351) | 2,580 (+130) | 2,681 (+231) | 2,755 (+305) | 113 (4% of 2,924) | 222 (7% of 3,107) | 413 (12% of 3,399) |
| Patient 6 | | | | 2,970 (+871) | 3087 (+117) | 3,082 (+112) | 3,481 (+511) | 116 (3% of 3,315) | 331 (8% of 3,929) | 148 (4% of 3,741) |
| Patient 7 | Nodular | | 2,504 (+744) | 2,942 (+438) | 3,156 (+214) | 3,290 (+348) | 3,026 (+84) | 180 (5% of 3,684) | 60 (2% of 3,300) | 258 (7% of 3,632) |
| Patient 8 | Nodular | | | 3,739 (+1,235) | 3,808 (+69) | 3,952 (+213) | 3,911 (+172) | 151 (4% of 4,172) | 89 (2% of 4,069) | 127 (3% of 4,251) |
| Patient 9 | | | | 3,486 (+982) | 3,649 (+163) | 3,693 (+207) | 3830 (+344) | 160 (4% of 4,016) | 129 (3% of 4,122) | 121 (3% of 4,158) |

**Fig 3. Group-intersection analysis of proteomic data heterogeneity.** The Venn diagram schematically depicts the inter-group interactions. The table below the figure presents the sizes of various groups of proteins per each patient. Interpretation example: 1760 proteins are shared among all 27 samples from all 9 patients. Additional 749 proteins, which is in total 2509 proteins are shared among all 9 samples from all 3 Infiltrative patients. Patient 1 has the Infiltrative subtype, and it has 3200 proteins shared among all its samples, which is 691 proteins in addition to the intersection of its sub-type. An additional 118 proteins are shared between the samples from Location 1 and Location 2, resulting in 3318 proteins shared between these two locations. Finally, Location 1 has 85 proteins uniquely identified in this location, which is about 2% of the total 3656 proteins in the sample from this location. Note, that some of these 85 proteins may be observed in a different patient.

Table 2. BCC subtype proteomic data heterogeneity.

| Protein Group Description | Infiltrative | Superficial | Nodular |
|---|---|---|---|
| Proteins that appear in each patient, in *all 3 locations* | 2509 | 2099 | 2504 |
| Proteins that appear in each patient, in *at least 1 location* | 3602 | 3301 | 3573 |
| Proteins that appear in each patient, in *1 or 2 locations only* | 130 | 104 | 107 |

proteins observed in each Nodular patient, but not in Superficial patients (Table 3). Moreover, proteins CYP2W1 (Cytochrome P450) and NTRK3 (Neurotrophic Receptor Tyrosine Kinase 3) were never observed in Infiltrative and Superficial samples, yet always identified in Nodular patients (Table 3).

Finally, high inter-patient heterogeneity in BCC proteomics can be also observed from PCA decomposition of protein data (Fig 4). Even though most samples of the same patient are located in proximity to each other (as expected), we do observe many cases when this is true only for the single sample.

## Transcriptomic analysis

To further explore the multisite heterogeneity of the human BCC, we performed a transcriptomic analysis. One patient with each of 3 histologically confirmed BCC subtypes (infiltrative, superficial, nodular) was selected for the transcriptomics analysis. The total transcriptomics dataset consisted of 9 samples (3 per patient). A total of 32,397 genes were identified. The gene count matrix is available at the GitHub repository (https://github.com/GolbergLab/BCC_Heterogeneity, hosted at GitHub (GitHub 2022).

Group-intersection heterogeneity analysis of the transcriptomics data (Fig 5) provides additional information on the intra-patient heterogeneity. The diagram shows that 20,099 genes ("all samples" slice) expressed in all BCC samples, which is about 62% of all 32,397 observed genes. On the per-patient level, we observed 26,976 genes ("all patient loci") shared among all samples of the infiltrative BCC subject, 22,068 among samples of superficial and 27,373 among samples of nodular subjects. The number of uniquely identified genes per each sample ranged from 253 to 1763 (1–6% of total genes measured in that sample), implying high potential heterogeneity of the BCC transcriptomic landscape.

Differential expression analysis yielded 156 genes with significant intra-patient heterogeneity (Group 2) for at least 2 of the 3 patients (p = 2.10e-152). GO-term analysis (**Supplements S4, S5 in** S1 File) of these genes shows statistical enrichment in cornification (GO:0070268, hypergeometric test p = 3.2e-40, corresponding to FDR of 4.3e-36) and keratinization (GO:0031424, hypergeometric test p = 3.2e-25, corresponding to FDR of 2.1e-21) and several additional biological processes, which is in agreement with previous BCC studies [37, 38].

## Study of RNA mutations

Somatic SNVs were studied by comparing the sequencing data of intra-patient locations under the assumption that the distribution of expressed alleles should be similar across skin cells of the same individual [39]. Following this approach, we found 22, 30, and 6 genome locations with JS distance above 0.6 for patients 1, 4 and 7 respectively (**Supplement S2 in** S1 File). For example, genome position 70452749 in chromosome X of patient 7 with JS score of 0.77 suggests a possible point mutation in the *DLG3* gene. *DLG3* was previously shown to have a role in oral squamous cell carcinoma [40], glioblastoma [41] and breast cancer [42].

**Table 3. Comparison of the different BCC sub-types.**

| Protein Group Description | Group size | Gene Names |
|---|---|---|
| Each patient in **Infiltrative** subtype, *at least 1 location* | 3602 | |
| Each patient in **Infiltrative** subtype, *at least 1 location* and do not appear in **Superficial** | 11 | 1. MYOZ1<br>2. TBC1D5<br>3. ITGAL<br>4. SOSTDC1<br>5. IKBIP<br>6. ARHGEF2<br>7. ZC3HAV1L<br>8. WRNIP1<br>9. NFYB<br>10. GZMK<br>11. FBXO30 |
| Each patient in **Infiltrative** subtype, *at least 1 location* and do not appear in **Nodular** | 6 | 1. VWA8<br>2. WAS<br>3. SRPX<br>4. SYCP1<br>5. TM9SF1<br>6. MACROD1 |
| Each patient in **Infiltrative** subtype, *at least 1 location* and do not appear in **other subtypes** | 0 | |
| Each patient in **Superficial** subtype, *at least 1 location* | 3301 | |
| Each patient in **Superficial** subtype, *at least 1 location* and do not appear in **Infiltrative** | 0 | |
| Each patient in **Superficial** subtype, *at least 1 location* and do not appear in **Nodular** | 3 | 1. LIPE<br>2. SRPX<br>3. APOF |
| Each patient in **Superficial** subtype, *at least 1 location* and do not appear in **other subtypes** | 0 | |
| Each patient in **Nodular** subtype, *at least 1 location* | 3573 | |
| Each patient in **Nodular** subtype, *at least 1 location* and do not appear in **Infiltrative** | 4 | 1. CYP2W1<br>2. TCEAL4<br>3. NTRK3<br>4. EIPR1 |
| Each patient in **Nodular** subtype, *at least 1 location* and do not appear in **Superficial** | 14 | 1. SULT1E1<br>2. PKP4<br>3. CYP2W1<br>4. NTRK3<br>5. YTHDC1<br>6. SOSTDC1<br>7. IKBIP<br>8. ARHGEF2<br>9. WRNIP1<br>10. FOXI3;<br>FOXI2<br>11. CLASP2<br>12. BRD3<br>13. FBXO30<br>14. GPN1 |
| Each patient in **Nodular** subtype, *at least 1 location* and do not appear in **other subtypes** | 2 | 1. CYP2W1<br>2. NTRK3 |

# Discussion

This study of transcriptome and proteome BCC heterogeneity suggests a more complex structure of cancer-related genes in the same lesion and between patients with similar diagnostics, further supporting previous reports on the tissue bulk [8, 9, 12]. Our finding on multisite

Sample Visualization based on 3-component PCA

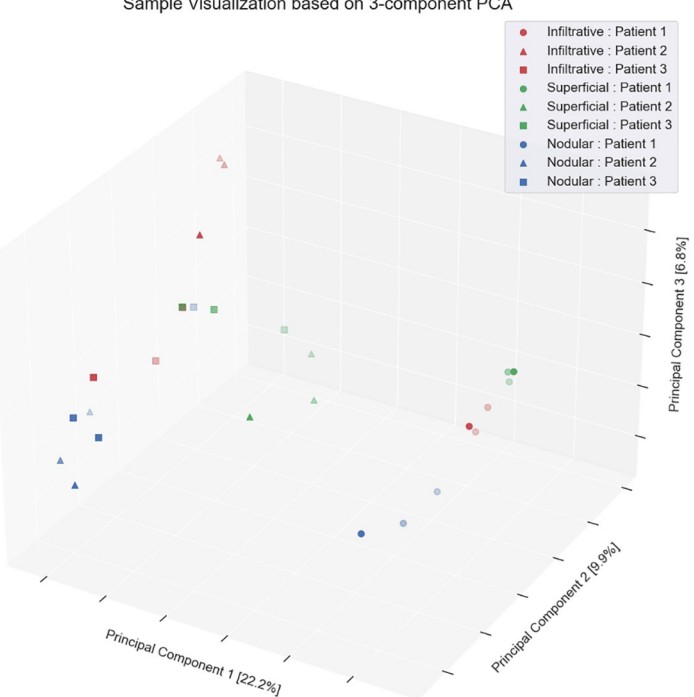

**Fig 4. 3D data representation based on PCA decomposition.** Red, green and blue marker colors correspond to samples from Infiltrative, Superficial and Nodular BCC sub-types respectively. Markers of the same color and shape correspond to samples obtained from the same patient.

sampling also further corroborate recent single cells studies on heterogeneity from the same site [43–45]. Differences identified in the RNA and proteome components of each unique BCC and between the different parts of each lesion may explain the different responses to pharmacological treatments [8, 12, 21, 24, 26, 27].

We observed the heterogeneous nature of BCC tumors when comparing RNA and proteins between 3 different sites extracted from a single lesion and proposed several approach to study inter- and intra- tumor heterogeneity.

To test the heterogeneity on the level of the entire genomic and proteomic profiles, we estimated and compared the Spearman and Pearson correlation values of Group 2 (correlations within the samples of the same patient) with the Group 1 (correlations within the samples of different patients) acting as a reference. The distribution of Group 2 correlations (Fig 2) consists of clearly higher values; yet, with a large overlap with Group 1. The latter implies that in many cases the difference between two samples originating from the same lesion of a single patient can be as large as the difference between samples excised from different patients. Furthermore, the correlations within both groups were significantly lower than the LC/MS/MS process noise control group, strengthening the hypothesis that the difference between samples was not due to noise.

To assess the heterogeneity among the observed expressed genes and proteins extracted from different samples, we analyzed the presence and absence of specific genes and proteins across these samples. The results, presented in Figs 3 and 5, demonstrate the existence of dozens to hundreds of unique genes and proteins that were identified in only 1 of 3 samples extracted from 1 patients' lesion. Moreover, we also observed a similar number of proteins uniquely absent from only 1 of 3 samples from 1 patient, suggesting high sub-clonality of each

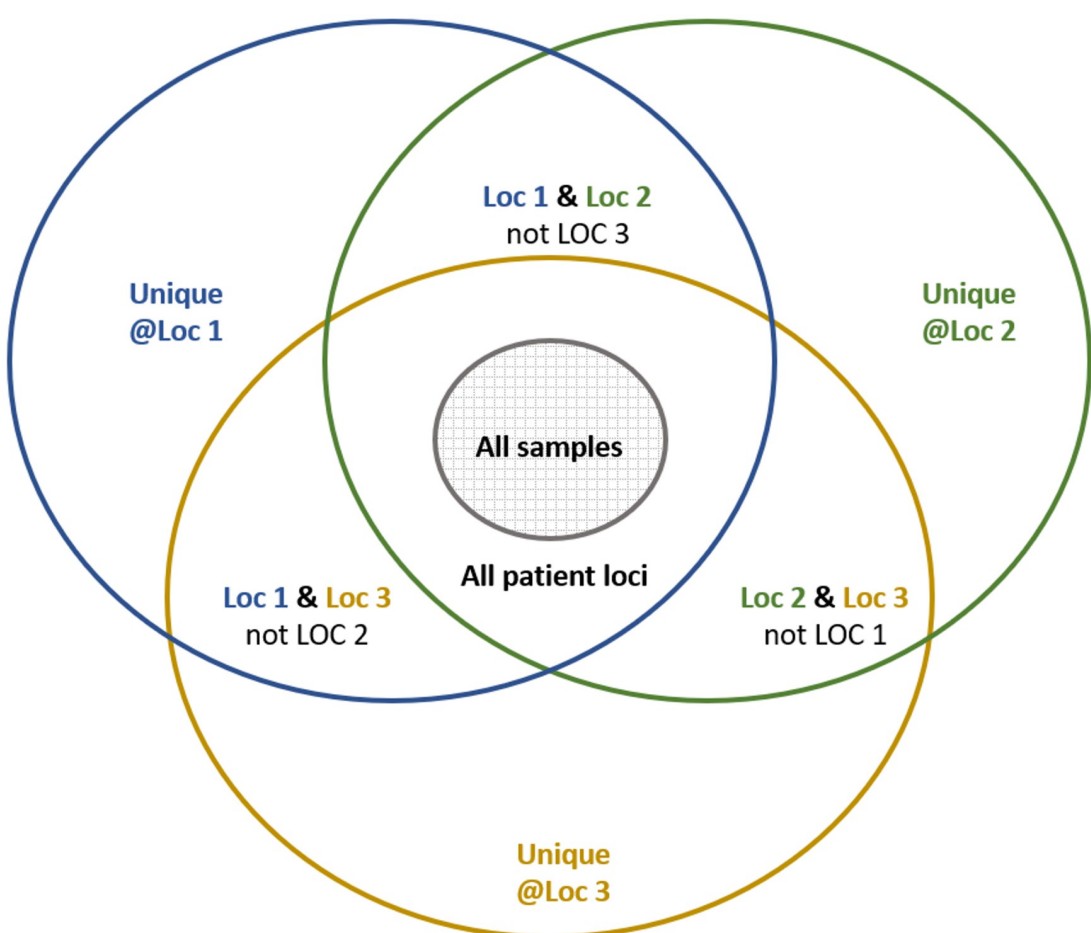

| | | Genes shared among | | | | | Genes unique to | | |
|---|---|---|---|---|---|---|---|---|---|
| | SubType | All samples | All patient loci | Loc 1 & Loc 2 not Loc 3 | Loc 1 & Loc 3 not Loc 2 | Loc 2 & Loc 3 not Loc 1 | Unique @Loc 1 | Unique @Loc 2 | Unique @Loc 3 |
| Patient 1 | Infiltrative | 20,099 | 26,976 (+6,877) | 29,113 (+2,137) | 28,818 (+752) | 27,945 (+969) | 368 (1% of 29,905) | 707 (2% of 30,789) | 203 (1% of 28,572) |
| Patient 4 | Superficial | | 22,068 (+1,969) | 23,940 (+1,872) | 22,711 (+643) | 24,268 (+2,200) | 654 (3% of 25,237) | 1,763 (6% of 27,903) | 956 (4% of 25,867) |
| Patient 7 | Nodular | | 27,373 (+7,274) | 28,558 (+1,185) | 28,492 (+1,119) | 27,776 (+403) | 669 (2% of 30,346) | 253 (1% of 29,214) | 506 (2% of 29,401) |

**Fig 5. Group-intersection analysis of RNA heterogeneity.** The Venn diagram schematically depicts the inter-group interactions. The table below the figure presents the sizes of various groups of genes per each patient. Interpretation example: 20,099 genes are shared among all 9 samples from all 3 patients. Patient 1 is of Infiltrative subtype, and it has 26,976 genes shared among all its samples, which is +6,877 genes more than the global intersection of all samples. An additional +2,137 genes are shared between the samples from Location 1 and Location 2, resulting in 29,113 genes shared between these two locations. Finally, Location 1 has 368 genes unique to this location, which is about 1% of all 29,905 genes observed at this location. Note, that some of these 368 genes may be identified in a different patient.

lesion. It is important to mention, that the analysis we done on the bulk tissue, thus, the ratio of tumor and healthy cells sampled may play a role in the observed heterogeneity.

Analysis of inter-subtype heterogeneity identified proteins which were observed in the different BCC subtypes but not in others, suggesting a high inter-sample heterogeneity.

The protein analysis yielded 2 proteins, CYP2W1 (Cytochrome P450) and NTRK3 (Neurotrophic Receptor Tyrosine Kinase 3) which were unique to the nodular BCC subtype.

CYP2W1 is a monooxygenase enzyme that has been shown to be expressed specifically in tumor tissues and during fetal life. A high expression of this enzyme was observed in colorectal cancers and its expression has been correlated with poor survival in a subset of colorectal cancer patients [43, 46]. This enzyme could be of prognostic and future therapeutic value NTRK gene fusions (encoding the neurotrophin receptors) are known oncogenic drivers of various tumors and the treatment of NTRK fusion-positive cancers includes the use of tyrosine kinase receptor inhibitors [44, 45]. A study by Dai et al. has shown this fusion gene to be upregulated in BCC when compared to normal skin [47].

Finally, on the single-gene level, we identified 156 genes that tend to be expressed heterogeneously in all BCC subtypes. Moreover, for 3 patients with transcriptomic data, we identified 6, 22 and 30 genome locations (SNVs) with high intra-patient variability in frequencies of the expressed nucleotides. In genome locations with expression of more than 2 alleles, these findings suggest the emergence of a somatic SNV. Alternative causes for the 2 heterozygous cases are that both copy number variations and mutations in one chromosome regulatory site can cause 1 of the alleles to be expressed more compared to the other allele.

Our results confirm the hypothesis that BCC tumors of infiltrative, superficial and nodular subtypes are heterogeneous and that different parts of the lesion express various molecular characteristics. This observation is in accordance with previous studies that demonstrated the heterogeneous nature of BCC tumors [26]. Moreover, BCC patients may present with several malignant skin lesions at the same time and the tumor heterogeneity observed in this study implies that lesions on the same patient might differ from those that are adjacent. This heterogeneity can have profound implications when considering targeted molecular therapy for BCC as this may be impeded by the inherent resistance of tumors. This conclusion is supported by reported cases of sporadic BCCs with lower response rates and increased therapy resistance [43–47].

The heterogeneous nature of tumors, as observed in our results, indicates that the location of the tissue analyzed from the biopsy specimen is of utmost importance. Thus, sampled by standard biopsies tissue does not present the full genetic profile of the tumor, but rather a small fraction of a complex genetic and molecular tumor topography. Thus, novel technologies that will support molecular sampling of multiple locations from a tumor during the same procedure are needed.

The findings from this study could provide the initial concept for research and development of novel diagnostic methods and novel targeted therapies informed by the genetic composition of several locations in a tumor.

## Conclusions

This study shows that intra-patient and inter-patient proteome and RNA expression variability are comparable, which implies the inherent heterogeneity of BCC lesions. The novel approach to the analysis of the transcriptome and proteome heterogeneity in a single BCC tumor presented here paves the way to better understanding of treatment resistance and tumor recurrence, which are the major limitations of cancer therapy. This knowledge will influence future research for diagnostic tools and will advance the development of personalized therapy.

## Supporting information

**S1 File. Methods, code and extended results.**
(DOCX)

## Acknowledgments

Israel Research Authority, Kamin program. Israel Ministry of Science and Technology, Euro-NanoMed3 MATISSE project, SPARK-TAU, TAU Zimin Institute for Engineering Solutions Advancing Better Lives.

We would like to acknowledge the surgeons at Meir Medical Center who assisted greatly with sample preparation and collection.

We also thank the Smoler Proteomics Center at the Technion, and especially Keren Benda-lak for help with the proteomic analysis.

We would also like to thank Tamar Lahav (Technion—Israel Institute of Technology) for useful discussions and suggestions regarding data analysis pipelines, and Amir Argoetti (Technion—Israel Institute of Technology) for fruitful discussions about the biology behind allele frequency differences in expression data.

## Author Contributions

**Conceptualization:** Ariel Berl, Ofir Shir-az, Ilai Genish, Alexander Golberg, Edward Vitkin, Zohar Yakhini, Avshalom Shalom.

**Data curation:** Alexander Golberg, Edward Vitkin, Zohar Yakhini.

**Formal analysis:** Ariel Berl, Ofir Shir-az, Ilai Genish, Hadas Biran, Vladimir Kravtsov, Debora Kidron, Alexander Golberg, Edward Vitkin, Zohar Yakhini.

**Funding acquisition:** Alexander Golberg, Avshalom Shalom.

**Investigation:** Ariel Berl, Ofir Shir-az, Ilai Genish, Hadas Biran, Din Mann, Amrita Singh, Julia Wise, Vladimir Kravtsov, Debora Kidron, Alexander Golberg, Edward Vitkin, Zohar Yakhini.

**Methodology:** Ariel Berl, Ofir Shir-az, Ilai Genish, Alexander Golberg.

**Software:** Edward Vitkin.

**Supervision:** Alexander Golberg, Edward Vitkin, Zohar Yakhini, Avshalom Shalom.

**Validation:** Vladimir Kravtsov, Debora Kidron.

**Writing – original draft:** Ariel Berl, Ofir Shir-az, Ilai Genish.

**Writing – review & editing:** Hadas Biran, Din Mann, Alexander Golberg, Edward Vitkin, Zohar Yakhini, Avshalom Shalom.

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
