## [Decision Letter · Decision Letter 0]

19 Jun 2023

PONE-D-23-02986Spatial proteome heterogeneity of human basal cell carcinomaPLOS ONE

Dear Dr. Golberg,

Thank you for submitting your manuscript to PLOS ONE. After careful consideration, we feel that it has merit but does not fully meet PLOS ONE’s publication criteria as it currently stands. Therefore, we invite you to submit a revised version of the manuscript that addresses the points raised during the review process.

We look forward to receiving your revised manuscript.

Kind regards,

Jesús Espinal-Enríquez

Academic Editor

PLOS ONE

“Israel Research Authority, Kamin program. Israel Ministry of Science and Technology, EuroNanoMed3 MATISSE project, SPARK-TAU, TAU Zimin Institute for Engineering Solutions Advancing Better Lives. “

“Israel Research Authority, Kamin Program- Golberg, Shalom, Yakhini.

Israel Ministry of Science and Technology- Golberg, Yakhini

EuroNanoMed3 MATISSE Project- Golberg, Shalom

SPARK-TAU- Golberg, Shalom, Yakhini

TAU Zimin Institute for Engineering Solutions Advancing Better Lives- Golberg.

Additional Editor Comments:

Dear Dr. Goldberg,

I write to you this letter to express my most sincere apologies. It has been unexpectedly difficult to obtain the revisions of both reviewers. I am aware that this long delay will affect several aspects of your research, as well as your colleagues' research. I have been pressing to the Reviewer #2 to send their revision as soon as possible.

In the meantime, I have activated the discussion with one revision. Please respond the comments and concerns raised by Reviewer 1 and send the revised version of your manuscript.

My best regards,

Jesús Espinal-Enríquez

Reviewers' comments:

Reviewer's Responses to Questions

**Comments to the Author**

1. Is the manuscript technically sound, and do the data support the conclusions?

Reviewer #1: Yes

2. Has the statistical analysis been performed appropriately and rigorously? 

Reviewer #1: No

3. Have the authors made all data underlying the findings in their manuscript fully available?

Reviewer #1: Yes

4. Is the manuscript presented in an intelligible fashion and written in standard English?

Reviewer #1: Yes

5. Review Comments to the Author

Reviewer #1: The authors have presented a molecular characterization of basal breast carcinoma and have discussed about spatial heterogeneity of its proteome. While their arguments are reasonable and their claims are somewhat expected, I have some doubts in relation to their statistical experimental design:

1. I cannot see any reason for such a restricted sample number. Even though their samples are apparently well characterized, using 9 samples for proteome analysis and 3 samples (!!) for transcriptomics is extremely restricted, even more if the main discussion is about heterogeneity. There is no attempt to calculate any measure of statistical power in their design. I suggest to consult with an expert statistician to improve on experimental design in order to have data that is actually able to support their assertions.

2. Along these same lines, it is not clear why if the authors already had 9 experimental samples, they used only 3 in the transcriptome analysis that is currently more straightforward to be performed in larger sample counts.

3. I cannot understand the rationale of including SNPs in the characterization of spatial protein heterogeneity.

In brief, the issue under investigation is interesting (though it would be worthy to resort to current experimental techniques in single cell and spatial proteomics), but the poor experimental design and the somewhat disconnected narrative gives the impression (most likely erroneous) that this is just a collection of unrelated experimental results.

6. PLOS authors have the option to publish the peer review history of their article (what does this mean?). If published, this will include your full peer review and any attached files.

Reviewer #1: **Yes: **Enrique Hernandez-Lemus

---

## [Author Response · Author response to Decision Letter 0]

17 Jul 2023

Response on Review Comments to the Author

2. Has the statistical analysis been performed appropriately and rigorously?

Reviewer #1: No

5. Review Comments to the Author

Reviewer #1: The authors have presented a molecular characterization of basal breast carcinoma and have discussed about spatial heterogeneity of its proteome. While their arguments are reasonable and their claims are somewhat expected, I have some doubts in relation to their statistical experimental design:

1. I cannot see any reason for such a restricted sample number. Even though their samples are apparently well characterized, using 9 samples for proteome analysis and 3 samples (!!) for transcriptomics is extremely restricted, even more if the main discussion is about heterogeneity. There is no attempt to calculate any measure of statistical power in their design. I suggest to consult with an expert statistician to improve on experimental design in order to have data that is actually able to support their assertions.

2. Along these same lines, it is not clear why if the authors already had 9 experimental samples, they used only 3 in the transcriptome analysis that is currently more straightforward to be performed in larger sample counts.

3. I cannot understand the rationale of including SNPs in the characterization of spatial protein heterogeneity.

In brief, the issue under investigation is interesting (though it would be worthy to resort to current experimental techniques in single cell and spatial proteomics), but the poor experimental design and the somewhat disconnected narrative gives the impression (most likely erroneous) that this is just a collection of unrelated experimental results.

We sincerely appreciate the time invested by the esteemed reviewer in reading our article and the thoughtful feedback provided. 

We are aware that 27 proteomic samples and 9 transcriptomic samples (3 from each patient) can appear somewhat limited. However, our main null hypothesis was tailored for a small sample set. Specifically, the null hypothesis was that BCC lesions are molecularly homogeneous. Thus, the careful comparative analysis of multiple samples collected from the same lesion even from a very few patients is enough to rebut it in favor of the alternative (that BCC lesions are not molecularly homogeneous). Therefore, we respectfully disagree with the reviewer on the statistical quality of the analysis.

Our approach to reject this null hypothesis was by checking for an even stronger alternative. We tested the overlap between the distributions of inter-sample correlations for intra-lesion and inter-patient samples. This overlap, if exists, would indicate cases with the distance between pair of samples from the same lesion as significant as pair of samples from two different patients.

Both for proteomics and for transcriptomics data we show that the distributions of intra-lesion and inter-patient correlations are highly overlapping, clearly rebutting the assumption of lesion homogeneity.

To better emphasis this, we refined (changes are marked in red here) the manuscript as following: 

“…the meaningful intersection between the histograms implies significant heterogeneity of specific tumors, clearly rebutting the hypothesis of lesion proteomic homogeneity. In other words, in notably many cases, the difference between two proteomic samples extracted from the same tumor of a single patient was as significant as the difference between two proteomic samples extracted from two different patients, sometimes even from two different BCC subtypes. Quantitively, the Bayesian probability of misclassifying the correlation (Group 1 vs. Group 2) was 40.7% and 48.1% for Spearman and Pearson tests, respectively.” 

for proteomics data and 

 “…the meaningful intersection between the histograms implies significant heterogeneity of specific tumors, clearly rebutting the hypothesis of lesion transcriptomic homogeneity. In other words, in some cases the difference between two RNA samples extracted from the same tumor of a single patient was as significant as the difference between two RNA samples extracted from two different patients with different BCC subtypes. Quantitively, the Bayesian probability of misclassifying the correlation (Group 1 vs. Group 2) was 22.2% and 11.1% for Spearman and Pearson tests, respectively.” 

for transcriptomics data. 

Addressing the #2 comment of the reviewer, the decision to perform transcriptomics for 9 out of 27 samples was received both due to project budget considerations and in view of results from the proteomic analysis, that clearly indicated the molecular heterogeneity.

We also want to explain the rationale of including SNPs to respond the comment #3 of the esteemed reviewer. The goal of this analysis was to address another aspect of molecular heterogeneity by checking the possibility to identify newly appeared somatic mutations inside a lesion. If the inherent nature of the lesion is homogeneous, we would not be able to find such mutations. Since this is strictly an intra-lesion analysis, addressing multiple samples collected from the same lesion is sufficient, akin to the approach reported in many single cell studies. Thus, the results are reported separately for each of 3 patients with available transcriptomic data:

“Somatic SNPs were studied by comparing the sequencing data of intra-patient locations under the assumption that the distribution of expressed alleles should be similar across skin cells of the same individual [35]. Following this approach, we found 22, 30, and 6 genome locations with JS distance above 0.6 for patients 1, 4 and 7 respectively (Supplement S2).”

We would like to express our sincere appreciation to the esteemed reviewer for their time and effort. 

We would also like to thank the esteemed editor for their efforts in reading the article and finding the reviewers. We have carefully studied the feedback and suggestions provided and refined the manuscript accordingly. 

We believe that the submitted manuscript provides a valuable contribution to the field. We are open to any further comments from the editor and remain dedicated to ensuring the quality and relevance of our work. Thank you for your ongoing support and guidance throughout this process.

---

## [Decision Letter · Decision Letter 1]

11 Aug 2023

PONE-D-23-02986R1Spatial proteome heterogeneity of human basal cell carcinomaPLOS ONE

Dear Dr. Golberg,

Thank you for submitting your manuscript to PLOS ONE. After careful consideration, we feel that it has merit but does not fully meet PLOS ONE’s publication criteria as it currently stands. Therefore, we invite you to submit a revised version of the manuscript that addresses the points raised during the review process.

We look forward to receiving your revised manuscript.

Kind regards,

Jesús Espinal-Enríquez

Academic Editor

PLOS ONE

Additional Editor Comments:

Dear Dr. Golberg.

Please respond all comments and concerns raised by both reviewers. Please note that the Reviewer #2 decided to reject this version of your manuscript. However, after analyzing all concerns, I consider that they can be addressed in the revised version.

Reviewers' comments:

Reviewer's Responses to Questions

**Comments to the Author**

1. If the authors have adequately addressed your comments raised in a previous round of review and you feel that this manuscript is now acceptable for publication, you may indicate that here to bypass the “Comments to the Author” section, enter your conflict of interest statement in the “Confidential to Editor” section, and submit your "Accept" recommendation.

Reviewer #2: (No Response)

2. Is the manuscript technically sound, and do the data support the conclusions?

Reviewer #2: No

3. Has the statistical analysis been performed appropriately and rigorously? 

Reviewer #2: No

4. Have the authors made all data underlying the findings in their manuscript fully available?

Reviewer #2: Yes

5. Is the manuscript presented in an intelligible fashion and written in standard English?

Reviewer #2: No

6. Review Comments to the Author

Reviewer #2: The authors describe a multisite characterization of transcriptomic and proteomic profiles in a limited number of basal cell carcinoma tissues isolated from human patients. They calculated hierarchical clustering and defined two relevant biological groups based on their omic similarities. They detected inter- and intra-tumor heterogeneity. In my opinion, this study is a proof of concept of a particular analytical strategy, but a biological assessment cannot be performed in its current form due to the limited number of data and, consequently, the basic analysis that can be performed on this type of data. I believe it is not suitable for publication in its present form.

Mayor concerns

The authors state there is not a widely used strategy to assess heterogeneity and this paper proposes several approaches to do so. I disagree with the authors, since there are several robust techniques that have been applied to BBC such as single cells, spatial proteomics and dedicated computational methods mainly based on artificial intelligence such as machine learning approaches as well as graph neural networks and mutidmensional integration of omics data. Some elegant examples are: Unravelling the landscape of skin cancer through single-cell transcriptomics (10.1016/j.tranon.2022.101557), Single-cell analysis of human basal cell carcinoma reveals novel regulators of tumor growth and the tumor microenvironment (10.1126/sciadv.abm79) and Integrated multi-omics reveals cellular and molecular interactions governing the invasive niche of basal cell carcinoma (10.1038/s41467-022-32670-w). As I have already mentioned, I believe this is a homemade approach to describe a complex phenomenon.

The title needs to be changed: in formal terms, spatial proteomics aims to characterize protein locations and their dynamics at the subcellular level. The evaluation presented in this study analyzed multisite gene expression profiles (3 different samples extracted from a single lesion), not formally a spatial characterization. In addition, the title only mentions proteomic analysis, but in some tumors the transcriptomic lanscape is also presented. Please modify the title to better describe what was performed.

In my opinion, at least for protein analysis, it would be better from a computational and interpretive strategy point of view to calculate the k-mean in addition to hierarchical clustering.

Show the adjusted p-values for the correlation matrix (Spearman and Pearson) and not just the R-squared.

The authors mentioned that they detected differences between: "two RNA samples extracted from the same tumor from the same patient" and " between two RNA samples extracted from two different patients with different BCC subtypes". In either of these cases, tumor purity is an important feature that could affect these results, mainly because the authors described that the tumor tissue collected included both tumor tissue and healthy margins. Please note this important histopathologic feature in the analysis.

In the RNA-seq analysis I guess the authors did not remove those genes that are not expressed in x% of the samples before normalization, that is the reason why they had a large number of RNAs. Please remove uninformative genes from your analysis before normalize raw count data.

I believe that this study cannot describe a biological landscape for the limited number of samples evaluated. In other approaches such as single cell, the evaluation of few patients could provide relevant information because thousands of single cells are evaluated, in this case there is only a bulk evaluation and therefore I believe there is not enough data to make biological interpretations.

I do not believe that "SNP" analysis provides any relevant information. First, I think the correct term is mutational analysis and therefore the authors identified mutations in coding regions, not SNPs. Secondly, I believe that although mutational data are useful for defining subclonal architecture, in this case there is no specific analysis to assess it, and thus any relevant data for the biological questions.

Show the dendograms of the hierarchical clustering analysis.

Indicate in the PCA analysis the % of variance explained by each component.

Improve figure quality

7. PLOS authors have the option to publish the peer review history of their article (what does this mean?). If published, this will include your full peer review and any attached files.

Reviewer #2: **Yes: **Sandra Romero-Cordoba

---

## [Author Response · Author response to Decision Letter 1]

14 Sep 2023

13 September 2023

Dear Editor,

We would like to express our sincere appreciation to the esteemed reviewer for their time and effort. 

We would also like to thank the esteemed editor for their efforts in reading the article and finding the reviewers. We have carefully studied the feedback and suggestions provided and refined the manuscript accordingly. 

We believe that the submitted manuscript provides a valuable contribution to the field. We are open to any further comments from the editor and remain dedicated to ensuring the quality and relevance of our work. Thank you for your ongoing support and guidance throughout this process.

Sincerely,

Alexander Golberg

Point by point response. 

Reviewer #2: The authors describe a multisite characterization of transcriptomic and proteomic profiles in a limited number of basal cell carcinoma tissues isolated from human patients. They calculated hierarchical clustering and defined two relevant biological groups based on their omic similarities. They detected inter- and intra-tumor heterogeneity. In my opinion, this study is a proof of concept of a particular analytical strategy, but a biological assessment cannot be performed in its current form due to the limited number of data and, consequently, the basic analysis that can be performed on this type of data. I believe it is not suitable for publication in its present form.

We would like to comment on this summary of the esteemed reviewer. In our research we did not perform any hierarchical clustering and neither did we search for biological groups based on sample omics similarities.

The patients were grouped by the pathologist’s diagnosis according to the detected BCC subtype. The samples were grouped according to their patient origin, thus defining intra-tumor group of samples. Our primary goal was to investigate the level of heterogeneity of molecular measurements inside each tumor. The inter-tumor heterogeneity was investigated as a reference group for the observed results.

The main research hypothesis was tailored for a small sample set, specifically, that BCC lesions are molecularly homogeneous. Thus, the careful comparative analysis of multiple samples collected from the same lesion even from a very few patients is enough to rebut it in favor of the alternative (that BCC lesions are not molecularly homogeneous). Such a result has implications on decision made from molecular data obtained from a single-site biopsy. 

Mayor concerns

The authors state there is not a widely used strategy to assess heterogeneity and this paper proposes several approaches to do so. I disagree with the authors, since there are several robust techniques that have been applied to BBC such as single cells, spatial proteomics and dedicated computational methods mainly based on artificial intelligence such as machine learning approaches as well as graph neural networks and mutidmensional integration of omics data. Some elegant examples are: Unravelling the landscape of skin cancer through single-cell transcriptomics (10.1016/j.tranon.2022.101557), Single-cell analysis of human basal cell carcinoma reveals novel regulators of tumor growth and the tumor microenvironment (10.1126/sciadv.abm79) and Integrated multi-omics reveals cellular and molecular interactions governing the invasive niche of basal cell carcinoma (10.1038/s41467-022-32670-w). As I have already mentioned, I believe this is a homemade approach to describe a complex phenomenon.

We are grateful for the reviewer for bringing these novel studies to our attention. We added these important recent references and updated the text. Our approach also deals, as mentioned by the reviewers with several sites in the same lesion, which is, indeed, different from the single cells works published, which looks at many cells in the same site. The number of different patients in our work and published single cells studies is similar suggesting similar needed statistical tools to describe inter-patients heterogeneity. 

The title needs to be changed: in formal terms, spatial proteomics aims to characterize protein locations and their dynamics at the subcellular level. The evaluation presented in this study analyzed multisite gene expression profiles (3 different samples extracted from a single lesion), not formally a spatial characterization. In addition, the title only mentions proteomic analysis, but in some tumors the transcriptomic lanscape is also presented. Please modify the title to better describe what was performed.

Amended as requested

In my opinion, at least for protein analysis, it would be better from a computational and interpretive strategy point of view to calculate the k-mean in addition to hierarchical clustering.

Again, no clustering was performed in this research. Patients and their samples were grouped according to histopathological diagnosis.

Show the adjusted p-values for the correlation matrix (Spearman and Pearson) and not just the R-squared.

We do not report the coefficient of determination (R-squared) in our analysis. We report correlation coefficients (Spearman-R and Pearson-R). 

Since the vector lengths are equal for all analysis pairs, the correlation coefficients are directly correlated with the p-values (i.e. higher the correlation coefficient – lower the corresponding p-value). Therefore, for visualization simplicity we decided to report correlation coefficients only.

The authors mentioned that they detected differences between: "two RNA samples extracted from the same tumor from the same patient" and " between two RNA samples extracted from two different patients with different BCC subtypes". In either of these cases, tumor purity is an important feature that could affect these results, mainly because the authors described that the tumor tissue collected included both tumor tissue and healthy margins. Please note this important histopathologic feature in the analysis.

We agree with the reviewer and added the sentence on the limitation of the sampling and analysing the bulk tissue in the discussion. 

In the RNA-seq analysis I guess the authors did not remove those genes that are not expressed in x% of the samples before normalization, that is the reason why they had a large number of RNAs. Please remove uninformative genes from your analysis before normalize raw count data.

We agree that removing lowly-expressed genes is a common and useful practice. However, since we wish to evaluate variability between different samples of a tumor, excluding genes that are only expressed in one sample would be counterproductive. We decided to remove genes with less than 0.5 CPM (counts per million) in all samples (this threshold is advised by [1] and [2]). This did not affect the differential analysis, since we only considered genes that were found in at least two patients, however intersection analysis slightly changed and was updated.

[1] Chen, Yunshun, Aaron TL Lun, and Gordon K. Smyth. "From reads to genes to pathways: differential expression analysis of RNA-Seq experiments using Rsubread and the edgeR quasi-likelihood pipeline." F1000Research 5 (2016).

[2] Galaxy training! “2: RNA-seq counts to genes”. URL: https://training.galaxyproject.org/training-material/topics/transcriptomics/tutorials/rna-seq-counts-to-genes/tutorial.html#filtering-to-remove-lowly-expressed-genes

I believe that this study cannot describe a biological landscape for the limited number of samples evaluated. In other approaches such as single cell, the evaluation of few patients could provide relevant information because thousands of single cells are evaluated, in this case there is only a bulk evaluation and therefore I believe there is not enough data to make biological interpretations.

The goal of this study was to show heterogeneity of the tumor in multiple sites with implications for diagnostics done with single site biopsy. We clarified this in the revised version. The biological interpretation shown is the inter- and intra patient heterogeneity. 

I do not believe that "SNP" analysis provides any relevant information. First, I think the correct term is mutational analysis and therefore the authors identified mutations in coding regions, not SNPs. Secondly, I believe that although mutational data are useful for defining subclonal architecture, in this case there is no specific analysis to assess it, and thus any relevant data for the biological questions.

We removed the use of “SNPs” and replaced it with “somatic SNVs” and “point mutations”. As to the second remark, we believe that this analysis helps in establishing the heterogeneity of BCC tumors. If the tumors are homogenous, the distribution of different nucleotides in a genomic location should be roughly the same between different samples of the same tumor. For example, let’s say that in one sample we see 2 different nucleotides in the same genomic location: 30 RNA strands with “A” and 70 RNA strands with “C”. These could be the two germline alleles of the individual. However, if this is the case, we would expect that in all three samples that belong to the same individual, the distribution of alleles would be roughly the same: 3:7 between “A”s and “C”s. However, if we see a significantly different distribution of “A”s and “C”s (or “G”s or “U”s) in the other samples, then a possible explanation would be that a mutation exists, and is more prevalent in one section of the tumor then in other sections. There are also other possible explanations, for example that there’s a copy number variation that affects this distribution or that another biological process affects the expression of germline alleles. However, all of these possible explanations rely on the fact that different samples of the same tumor behave differently (whether due to a point mutation or for another reason), so they all contribute to the heterogeneity explanation. 

Show the dendograms of the hierarchical clustering analysis.

No clustering was performed in this research.

Indicate in the PCA analysis the % of variance explained by each component.

Amended as requested.

Improve figure quality

Amended as requested.

---

## [Decision Letter · Decision Letter 2]

19 Oct 2023

Exploring Multisite Heterogeneity of Human Basal Cell Carcinoma

Proteome and Transcriptome

PONE-D-23-02986R2

Dear Dr. Golberg,

We’re pleased to inform you that your manuscript has been judged scientifically suitable for publication and will be formally accepted for publication once it meets all outstanding technical requirements.

Kind regards,

Jesús Espinal-Enríquez

Academic Editor

PLOS ONE

Additional Editor Comments (optional):

Dear authors.

All the comments and concerns have been addressed in this new version. The manuscript has improved, and it is more readable for a broad audience.

Reviewers' comments:

Reviewer's Responses to Questions

**Comments to the Author**

1. If the authors have adequately addressed your comments raised in a previous round of review and you feel that this manuscript is now acceptable for publication, you may indicate that here to bypass the “Comments to the Author” section, enter your conflict of interest statement in the “Confidential to Editor” section, and submit your "Accept" recommendation.

Reviewer #1: All comments have been addressed

2. Is the manuscript technically sound, and do the data support the conclusions?

Reviewer #1: Yes

3. Has the statistical analysis been performed appropriately and rigorously? 

Reviewer #1: Yes

4. Have the authors made all data underlying the findings in their manuscript fully available?

Reviewer #1: Yes

5. Is the manuscript presented in an intelligible fashion and written in standard English?

Reviewer #1: Yes

6. Review Comments to the Author

Reviewer #1: The manuscript is now clearer and the findings are potentially useful to other researchers in the field.

7. PLOS authors have the option to publish the peer review history of their article (what does this mean?). If published, this will include your full peer review and any attached files.

Reviewer #1: **Yes: **Enrique Hernandez-Lemus

---

## [Editor Report · Acceptance letter]

24 Oct 2023

PONE-D-23-02986R2 

Exploring Multisite Heterogeneity of Human Basal Cell Carcinoma Proteome and Transcriptome.  

Dear Dr. Golberg:

I'm pleased to inform you that your manuscript has been deemed suitable for publication in PLOS ONE. Congratulations! Your manuscript is now with our production department. 

Kind regards, 

on behalf of

Dr. Jesús Espinal-Enríquez 

Academic Editor

PLOS ONE